

# Actual physics behind mono-X

Elias Bernreuther, Jan Horak, Tilman Plehn and Anja Butter

Institut für Theoretische Physik, Universität Heidelberg, Germany

## Abstract

Mono-X searches are standard dark matter search strategies at the LHC. First, we show how in the case of initial state radiation they essentially collapse to mono-jet searches. Second, we systematically study mono-X signatures from decays of heavier dark matter states. Direct detection constraints strongly limit our MSSM expectations, but largely vanish for mono-Z and mono-Higgs signals once we include light NMSSM mediators. Finally, the decay topology motivates mono-W-pair and mono-Higgs-pair searches, strengthening and complementing their mono-X counterparts.

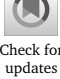

## Content

# 1  Introduction

Dark matter is one of the great puzzles in fundamental physics [1], with overwhelming evidence for an explanation in terms of new particles in the mass range between axions and primordial black holes. As a research field, it is driven by rapid experimental progress supporting many different search strategies, making it likely that the current and next generations of experiments will provide a definitive answer to several of the fundamental concepts and models.

No matter what physics hypothesis we base our dark matter searches on, we need to ensure that from a quantum field theory perspective the model makes sense [2, 3, 7]. This applies to an effective theory approach [10, 12], leading order Feynman diagrams nowadays called simplified models [14], and actual UV completions of the Standard Model [18, 25, 29]. Among the candidates for the latter, supersymmetry [30, 35] still stands out for two reasons: first, it offers a dark matter model as part of a perturbative gauge theory which can be evolved to fundamental scales; second, aside from a theoretical fine-tuning argument hardly related to the dark matter sector, it does not lose its attractive features once we include the current LHC constraints [37].

Dark matter searches at colliders have a long history. Invisible particles recoiling against visible particles have been searched for at least since the UA1/UA2 days. The corresponding visible particles can in principle be jets, leptons, photons, weak bosons, or Higgs bosons. The corresponding searches have been dubbed mono-$X$ searches and are often motivated through effective theory arguments. We take the opposite approach and classify many of the viable mono-$X$ searches through two event topologies:

1. dark matter mediators in the hard process, recoiling against Standard Model particles from *initial state radiation*;

2. production of heavier dark sector states followed by *dark matter decays* to Standard Model particles and the actual dark matter agent [38].

Dark matter combined with initial state radiation (ISR) of Standard Model particles allows for a systematic comparison of different mono-$X$ channels [39]. In Sec. 2 we will quantitatively compare mono-jet [40], mono-photon [44], and mono-$Z$ [46] production for an on-shell $Z'$-mediator at the LHC. Aside from the obvious question which mono-$X$ channel works best, we will also ask what we learn from combining different such mono-$X$ signatures. Other mediators, for example including (pseudo-)scalars coupling to gluons, prefer mono-jet signatures by construction. Our findings can easily be generalized to a proper $2 \rightarrow 3$ process, except that in this case the LHC mono-$X$ rate will be negligibly small.

The second topology is motivated for example by supersymmetric electroweakinos. They describe dark matter as a combination of singlet, doublet, and triplet representations of $SU(2)_L$ and therefore include additional neutral and charged dark matter particles. Decays of heavy neutralinos and charginos to the lightest neutralino allow us to compare mono-$Z$ [46], mono-$W$ [52], mono-Higgs [55] signatures in Sec. 4. A side aspect of this classification is that invisible $Z$-decays and invisible Higgs decays are naturally included in our approach. We decouple the heavy Higgs mediators, which are already established as the motivation for mono-$Z$ and mono-Higgs signatures [25].

For a proper dark matter model, the combination of relic density and direct detection constraints strongly cuts into the LHC signals, especially for the mono-$Z$ case. One way to avoid direct detection constraints is to rely more on heavier neutralinos and charginos. This leads us to consider mono-$W$-pair and mono-Higgs-pair signatures. We find that in contrast to

the usual effective theory scenarios, decay topologies prefer mono-*W* over mono-*Z* signatures, where the former include sizable contributions from mono-*W*-pairs. The most flexible process in avoiding current constraints is mono-Higgs-pairs, where already in the MSSM we can largely decouple the LHC and direct detection processes.

Finally, in Sec. 5 we expand the mediator sector and study the singlet–singlino dark matter sector in the NMSSM. This effectively decouples the relic density constraint from our LHC analysis. We focus on the most constrained mono-*Z* and the most flexible mono-Higgs-pair signatures and show how the light scalar and pseudoscalar mediators allow us to avoid the corner which the relic density and direct detection constraints usually push us into. Instead, we find sizable signal rates for the LHC in the presence of all available constraints.

## 2   Initial state X-radiation

If we assume that the dark matter mediator couples to quarks, a universal topology of dark matter signatures is given by initial state radiation (ISR) of a gluon, a photon, or a *Z*-boson, shown in Fig. 1. To illustrate their main features we employ a model with a heavy $Z'$ mediator combined with a Majorana fermion $\chi$ as a dark matter candidate. For our toy model the mediator couples to the incoming quarks and to the dark matter particles and can, if heavy, be integrated out. The signal process then reads

$$pp \to Z'X \to \chi\chi X \qquad \text{with} \quad X = j, \gamma, Z \,. \tag{1}$$

For the mono-*Z* signal we need to include a decay. While hadronic decays $Z \to q\bar{q}$ come with a large branching ratio, leptonic decays like $Z \to \ell\ell$ can help experimentally. Mono-*W* events can occur through ISR when we use a $q\bar{q}'$ initial state to generate a hard $q\bar{q}$ scattering. Finally, mono-Higgs signatures obviously make no sense when we rely on ISR.

Our toy model benefits from the phase space enhancement of an on-shell $Z'$ mediator in the *s*-channel of the hard process, but the hard process can be easily replaced by any other *s*-channel or *t*-channel mediator exchange [12]. In that case the challenge will be to enhance the cross section to explain the observed relic density [1] and predict a visible LHC signal for a perturbative and predictive quantum theory. We only use our toy model to illustrate the different mono-*X* channels, for a discussion of possible UV completions including a $Z'$ mediator we refer to the detailed discussion in Ref. [29].

From the similarity of the three Feynman diagrams in Fig. 1 we first derive that in the limit $m_Z \ll m_{Z'}$ the total rates for the different mono-*X* processes scale like

$$\frac{\sigma_{\chi\chi\gamma}}{\sigma_{\chi\chi j}} \approx \frac{\alpha}{\alpha_s} \frac{Q_q^2}{C_F} \approx \frac{1}{40}$$

$$\frac{\sigma_{\chi\chi\ell\ell}}{\sigma_{\chi\chi j}} \approx \frac{\alpha}{\alpha_s} \frac{Q_q^2 s_w^2}{C_F} \text{BR}(Z \to \ell^+\ell^-) \approx \frac{1}{2000} \,. \tag{2}$$

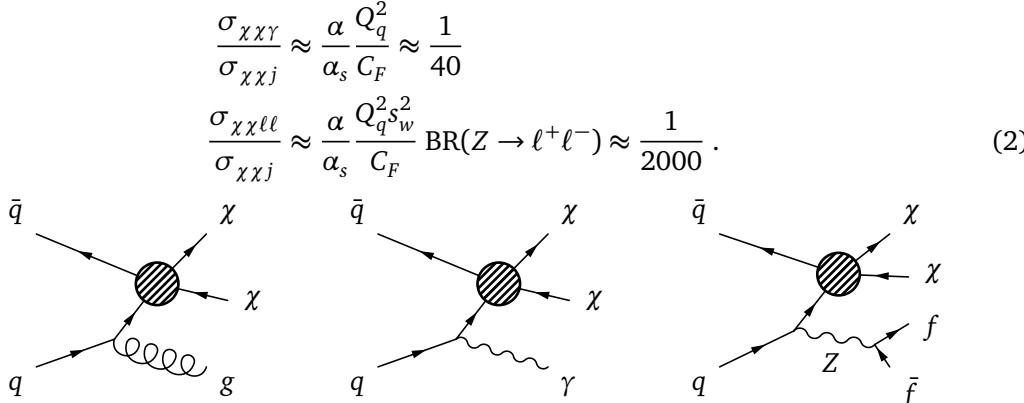

Figure 1: Feynman diagrams contributing to mono-*X* production

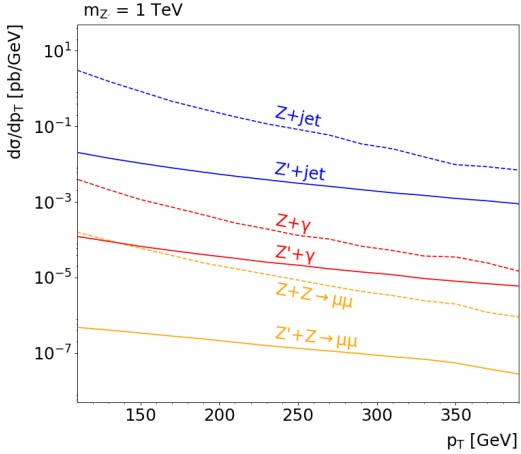
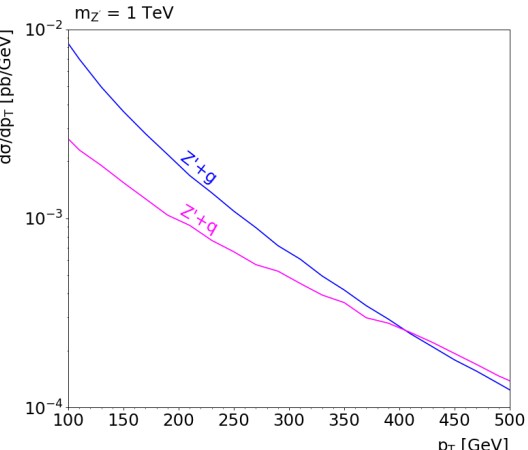

Figure 2: Transverse momentum spectrum for different mono-$X$ signals and backgrounds assuming a heavy vector mediator (left). The spectrum for the jet is composed by a $q\bar{q}$ initial state with a final state gluon and a quark gluon initial state with a quark jet radiated (right), here decomposed into fractions of the total signal.

Once we include the $Z$-mass the actual suppression of the mono-$Z$ channel is closer to $10^{-4}$. In addition, the Feynman diagrams also suggest that any kinematic $x$-distribution scales like

$$\frac{1}{\sigma_{\chi\chi j}} \frac{d\sigma_{\chi\chi j}}{dx} \approx \frac{1}{\sigma_{\chi\chi\gamma}} \frac{d\sigma_{\chi\chi\gamma}}{dx} \approx \frac{1}{\sigma_{\chi\chi ff}} \frac{d\sigma_{\chi\chi ff}}{dx} . \tag{3}$$

We show the $p_T$ distributions for the different mono-$X$ channels in Fig. 2, indicating that there is indeed no visible difference between their shapes. Once we include phase space information, the suppression of the mono-photon becomes stronger, because the rapidity coverage of the detector for jets extends to $|\eta| < 4.5$, while photons rely on an efficient electromagnetic calorimeter with $|\eta| < 2.5$. On the other hand, photons can be detected to significantly smaller transverse momenta than jets.

Finally, the topology shown in Fig. 1 is not complete for mono-jet production. In this case the gluon can be crossed to the initial state, as shown in Fig. 3. The size of this correction for dark matter mediator radiation off quarks can be sizable and depends on the transverse momentum of the jet, as shown in the right panel of Fig. 2. On the other hand, the mediator could also couple to incoming gluons and this way produce essentially only gluon jets. Distinguishing these two mono-jet hypotheses is a perfect case for including quark-gluon discrimination by default in any mono-jet analysis.

The same scaling arguments as for the mono-$X$ signal apply to the leading background,

$$pp \rightarrow Z_{\nu\nu}X \qquad \text{with} \quad X = j, \gamma, Z , \tag{4}$$

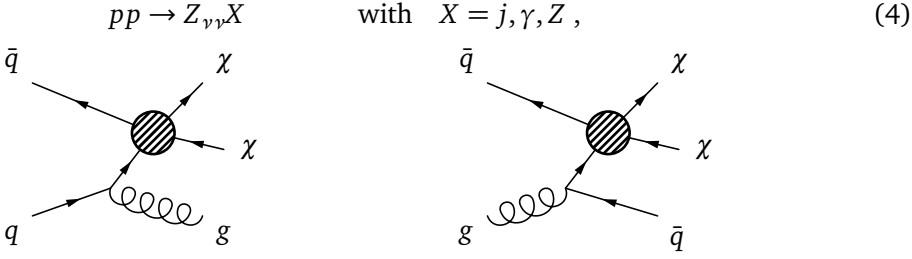

Figure 3: Feynman diagrams contributing to mono-jet production.

possibly with the exception of mono-$Z$ production, where the hard process and the collinear radiation are now both described by $Z$-production. This means that the signal scaling of Eq.(2) also applies to the leading backgrounds. In the left panel of Fig. 2 we see that indeed all mono-$X$ background scale similarly, and all background are slightly softer than the corresponding signals. This is an effect of the logarithmic collinear enhancement with the $Z'$ mass.

If our discovery channel is statistics limited, the significances $n_\sigma$ for the different channels are given in terms of the luminosity, efficiencies, and the cross sections

$$n_{\sigma,j} = \sqrt{\epsilon_j \mathcal{L}} \, \frac{\sigma_{\chi\chi j}}{\sqrt{\sigma_{\nu\nu j}}} \qquad \Rightarrow \qquad n_{\sigma,\gamma} = \sqrt{\epsilon_\gamma \mathcal{L}} \, \frac{\sigma_{\chi\chi\gamma}}{\sqrt{\sigma_{\nu\nu\gamma}}} \approx \frac{1}{6.3} \sqrt{\frac{\epsilon_\gamma}{\epsilon_j}} \, n_{\sigma,j} \,. \tag{5}$$

Unless the efficiency correction factors, including acceptance cuts and cuts rejecting other backgrounds, point towards a very significant advantage for the mono-photon channel, the mono-jet channel will be the most promising search strategy. Using the same argument, the expected mono-$Z$ significance will be negligible. Note that the main assumption behind this estimate is that the leading uncertainties are statistical. For example with a jet energy scale uncertainty in ATLAS around 1 ... 2%, not far from the muon energy scale uncertainty, this is definitely the case.

An interesting aspect occurs when we include hadronic decays in mono-$Z$ production [60]. In that case, the two jets neither guarantee trigger, nor are they particularly useful to suppress backgrounds. Instead, the boosted $Z$-boson contributes to the mono-jet rate. We can estimate the boost necessary to identify the $Z$ as one jet from the separation of the two decay jets with the momentum fractions $z$ and $1-z$,

$$\Delta R_{jj} \approx \frac{m_Z}{p_{T,Z}} \, \frac{1}{\sqrt{z(1-z)}} > \frac{m_Z}{2p_{T,Z}} \,. \tag{6}$$

This typical hyperbolic shape implies that $Z$-decays with for instance $\Delta R_{jj} < 0.5$ are sensitive to events with $p_{T,Z} > m_Z$. While this is an interesting observation, it does not help with the ISR signature, because even including the hadronic $Z$-decays the universal mono-jet rate above the same threshold is around 200 times larger.

Given the impressive control of ATLAS and CMS over their systematic uncertainties it appears obvious, that ISR is not a valid justification to search for dark matter production using the mono-photon or mono-$Z$ signatures. Universally, mono-jet searches will always be much more powerful. This is even more obvious when we extend our models from the tree-level $Z'$ vector mediator to other simplified or full models [12]. For example scalar mediators in the $s$-channel preferably couple to gluons and clearly prefer mono-jet searches; scalar, color-charged mediators in the $t$-channel can always be produces on-shell and predict mono-jet events from ISR as well as from on-shell production with a subsequent decay. The link between mono-$X$ searches and such decays will be the subject of the next section.

Not even the argument that we would like to study the properties of dark matter by combining different LHC mono-$X$ channels holds in our case. All we can learn from mono-photon and mono-$Z$ signals from ISR topologies is collinear radiation of Standard Model particles off hard incoming quarks.[1]

---

[1]While we are aware that this negative bottom line is known to many, we took the opportunity to illustrate it quantitatively as an introductory part of our study.

# 3 Non-minimal dark matter sectors

A much more promising topology leading to mono-*X* signatures appears for non-minimal dark matter sectors. They consist of a dark matter agent, a SM or new physics mediator, and additional dark matter states. Such degrees of freedom arise when we embed dark matter in $SU(2)_L$ multiplets. The best-known example is the MSSM with mixed bino, wino, and higgsino dark matter, plus a pair of Dirac charginos. The electroweakino interactions reflect the supersymmetry of the Lagrangian. Especially on the mediator side we gain additional freedom from the NMSSM with its extended scalar sector.

Note that this approach is exactly the opposite to the usual effective field theory approaches, because it requires us to consider these additional particles as propagating degrees of freedom at the LHC and in the early universe.

## 3.1 MSSM

The minimal supersymmetric electroweakino sector [30] combines the fermionic partners of the weak gauge bosons and two Higgs doublets. The corresponding mass matrix is

$$
\begin{pmatrix}
M_1 & 0 & -m_Z c_\beta s_w & m_Z s_\beta s_w \\
0 & M_2 & m_Z c_\beta c_w & -m_Z s_\beta c_w \\
-m_Z c_\beta s_w & m_Z c_\beta c_w & 0 & -\mu \\
m_Z s_\beta s_w & -m_Z s_\beta c_w & -\mu & 0
\end{pmatrix} .
\tag{7}
$$

It is diagonalized through an orthogonal transformation $N$. Two higgsino doublets are not only required by the supersymmetric nature of the MSSM Higgs sector, they also generally ensure that higgsino loops do not lead to anomalies. The annihilation process, which guarantees the observed relic density in the MSSM, proceeds through a set of mediators, namely the $Z$ and Higgs bosons of the Standard Model, heavy new Higgs bosons, or new scalar partners of the Standard Model fermions [61]:

- $Z$-funnel annihilation through the higgsino components,

$$
g_{Z\tilde{\chi}_i^0\tilde{\chi}_j^0} = \frac{g}{2c_w}\left(N_{i3}N_{j3} - N_{i4}N_{j4}\right) .
\tag{8}
$$

  This coupling vanishes in the limit $t_\beta \to 1$ with equal higgsino fractions. Because the axial-vector component does not have a velocity suppression, the annihilation rate $\langle\sigma v\rangle$ prefers neutralino masses slightly above or below 45 GeV; directly on the $Z$-pole the annihilation is too efficient;

- light $h$-funnel annihilation, where the dark matter mass is around $m_{\tilde{\chi}_1^0} = m_h/2$ GeV, slightly away from the resonance. The underlying coupling

$$
g_{h\tilde{\chi}_i^0\tilde{\chi}_j^0} = \frac{1}{2}\left(g' N_{i1} - g N_{i2}\right)\left(s_\alpha N_{j3} + c_\alpha N_{j4}\right) + (i \leftrightarrow j)
\tag{9}
$$

  relies on higgsino-gaugino mixing. The angle $\alpha$ rotates the scalar Higgses into mass eigenstates. Given the SM-like nature of the light MSSM Higgs, almost the entire neutralino annihilation rate through the light Higgs funnel goes to $b\bar{b}$. The coupling then has the approximate form

$$
g_{h\tilde{\chi}_i^0\tilde{\chi}_j^0} \approx \frac{1}{2}\left(g' N_{i1} - g N_{i2}\right) s_\beta\left(-\frac{N_{j3}}{t_\beta} + N_{j4}\right) + (i \leftrightarrow j) ;
\tag{10}
$$

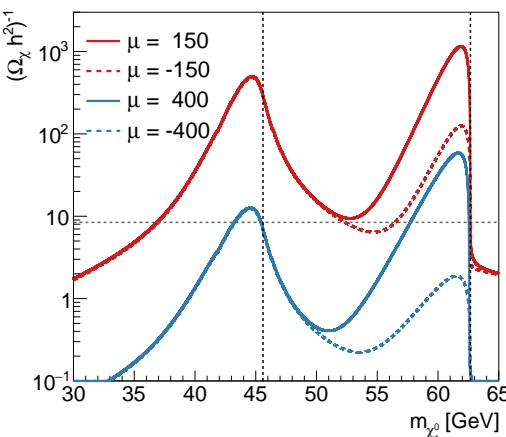

Figure 4: Inverse relic density near the $Z$-pole and Higgs pole in the MSSM.

- heavy Higgs funnel annihilation, where the pseudoscalar $A$ leads to efficient $s$-wave annihilation. The coupling is again driven by higgsino-gaugino mixing. Heavy scalar decays to down-type fermions are enhanced by $t_\beta$, which implies that for $t_\beta \gtrsim 30$ the resonance pole gets washed out and a $b\bar{b}$ final state appears;

- $t$-channel chargino exchange $\tilde{\chi}_1^0 \tilde{\chi}_1^0 \to WW$, relying on the coupling

$$g_{W\tilde{\chi}_i^0\tilde{\chi}_j^+} = g\left(\frac{1}{\sqrt{2}}N_{i4}V_{j2}^* - N_{i2}V_{j1}^*\right);\tag{11}$$

- $t$-channel neutralino exchange, $\tilde{\chi}_1^0 \tilde{\chi}_1^0 \to ZZ$ or $\tilde{\chi}_1^0 \tilde{\chi}_1^0 \to hh$. For the annihilation into $Z$ pairs, the relevant axial-vector coupling is illustrated in Eq.(8). For the annihilation to Higgs pairs, the relevant scalar coupling involves a product of higgsino and gaugino fractions, as given in Eq.(9);

- $t$-channel sfermion exchange, e.g. tau sleptons. In this case, significant coupling requires a large wino fraction, which typically leads to excessively large annihilation into $W$ bosons for dark matter masses below around 1 TeV;

- co-annihilation channels are efficient whenever there is an additional supersymmetric particle within about 10% of the dark matter mass [63,66,68]. Additional light charginos or sfermions are strongly disfavored by LEP [69]. Within the electroweakino sector, co-annihilation significantly contribute for example for processes with a light chargino in the $t$-channel.

In Fig. 4 we illustrate the pole annihilation through SM-like mediators in the MSSM for $M_1 = 10 \dots 80$ GeV, $\tan\beta = 10$, and a decoupled wino. We see that because of the velocity distribution the dark matter mass should actually be slightly below the actual pole condition. While the velocity distribution is also responsible for the width of the Higgs pole, the width of the $Z$-pole is given by the physical $Z$-width. As discussed above, the neutralino coupling to the $Z$-mediator only involves the higgsino fraction and is therefore independent of the sign of $\mu$ or the exchange of the two entries of $\mu$ in the neutralino mass matrix. On the other hand, the gaugino-higgsino coupling to the SM-like Higgs is significantly larger for $\mu > 0$.

On the side of the dark matter agent the MSSM ansatz is extremely flexible, describing dark matter masses from tens of GeV to few TeV [35]. For the three pure states, the neutralino masses are fixed by the relic density prediction based on each $SU(2)_L$ representation. Wino

dark matter, (co-)annihilating to weak bosons has to be in the mass range of 2 TeV to 3 TeV. Higgsino dark matter, annihilates slightly less efficiently, has to have a mass between 1 TeV and 2 TeV. Pure bino dark matter is only feasible with very light sleptons in the $t$-channel. One we include mixing, the MSSM can explain the observed relic density over the entire mass range. The limitation of the MSSM is the number of mediators, especially given the LHC constraints on heavy Higgs bosons and $t$-channel sfermions. For dark matter at mass accessible to the LHC, we essentially have to rely on the SM, or SM-like $Z$ and Higgs mediators.

## 3.2 NMSSM

The NMSSM extension [78,81] not only adds another singlino to the neutralino sector, it also predicts a singlet extension of the scalar mediator sector. The additional model parameters in the scalar sector include

$$\{\lambda, A_\lambda, \kappa, A_\kappa\} \tag{12}$$

with the convenient combinations

$$\tilde{\kappa} = \frac{\kappa}{\lambda} \qquad \text{(singlino–higgsino mass ratio)}$$

$$\tilde{\lambda} = \frac{\lambda}{g} \qquad \text{(singlino–higgsino mixing)} . \tag{13}$$

The singlet mass entry in the extended Higgs mass matrix is given by

$$s_{2\beta} \frac{\tilde{\lambda}^2 A_\lambda}{2\mu} + \frac{\tilde{\kappa}\mu}{m_Z^2} \left(A_\kappa + 4\tilde{\kappa}\mu\right) . \tag{14}$$

The light scalar and pseudoscalar mediators from the new singlet allow for very efficient dark matter annihilation and then predicts a whole range of signatures at the LHC, including invisible Higgs decays [81]. For our analysis we will set

$$A_\lambda = 2\mu \left(\frac{1}{s_{2\beta}} - \tilde{\kappa}\right) \tag{15}$$

at the relevant scale, to decouple the singlet sector from the SM-like Higgs boson and avoid for example constraints from Higgs coupling strengths.

The neutralino in the NMSSM mass matrix has the same form as for the MSSM, extended by an additional singlino

$$\begin{pmatrix} M_1 & 0 & -m_Z c_\beta s_w & m_Z s_\beta s_w & 0 \\ 0 & M_2 & m_Z c_\beta c_w & -m_Z s_\beta c_w & 0 \\ -m_Z c_\beta s_w & m_Z c_\beta c_w & 0 & -\mu & -m_Z s_\beta \tilde{\lambda} \\ m_Z s_\beta s_w & -m_Z s_\beta c_w & -\mu & 0 & -m_Z c_\beta \tilde{\lambda} \\ 0 & 0 & -m_Z s_\beta \tilde{\lambda} & -m_Z c_\beta \tilde{\lambda} & 2\tilde{\kappa}\mu \end{pmatrix} . \tag{16}$$

The annihilation process to two SM fermions through a light scalar or pseudo-scalar mediator

$$\tilde{\chi}_1^0 \tilde{\chi}_1^0 \to a_s, h_s \to f\bar{f} \tag{17}$$

is, in the limit of one SM-like Higgs boson, mediated by the coupling

$$g_{a_s \tilde{\chi}_1^0 \tilde{\chi}_1^0} \approx g_{h_s \tilde{\chi}_1^0 \tilde{\chi}_1^0} \approx \lambda \sqrt{2} \left(N_{13} N_{14} - \tilde{\kappa} N_{15}^2\right) , \tag{18}$$

It allows for an efficient annihilation of much lighter dark matter for a properly adjusted mediator mass. Going all the way to dark matter with masses in the GeV range essentially avoids the Xenon-based direct detection (DD) constraints and leaves us with CMB and nucleosynthesis constraints.

For extended dark matter sectors with charged particles a key constraint comes from LEP. To be safe, we assume that any charged particle which can hence be pair-produced in $e^+e^-$ collisions and which decays to leptons, jet, photons, or missing energy has to be heavier than 103 GeV. This includes the light chargino, which through its wino or higgsino mass parameters is closely tied to some of the neutralinos. Only the bino in the MSSM and the NMSSM, and the singlino in the NMSSM can be lighter.

Finally, it is obviously possible to use mono-$X$ searches, or following the previous discussion mono-jet searches, to target electroweakinos. However, over most of the MSSM parameter space this leads to proper $2 \to 3$ production processes, with the corresponding phase-space suppression. The supersymmetric equivalent to the $2 \to 2$ topologies discussed in Sec. 2 would be searches for invisible $Z$ or Higgs decays, which will be part of the discussion in the following sections.

### 3.3 SFitter setup

Our analysis of the MSSM and the NMSSM is based on the SFITTER framework [61, 81, 82]. Because we focus on the MSSM and NMSSM electroweakinos, we decouple all scalar particles at 5 TeV, except for the SM-like Higgs and, in the NMSSM case, the light set of scalar and pseudo-scalar mediators. This includes the heavy 2HDM states, which can in principle play an important role for dark matter annihilation [25]. The light Higgs mass is adjusted to match the measured value $m_h = 125$ GeV with the help of $\tan\beta \equiv t_\beta$ and $A_t$ [85]. All observables included in our global analysis are listed in Tab. 1. We emphasize that for any well-defined model the observed relic density is a crucial experimental constraint. While searches for invisibly decaying particles at the LHC certainly do not have to be related to this observable, any more global interpretation in terms of dark matter will break down unless we include a valid dark matter production mechanism and the relic density constraint.

An interesting coincidence appears when we compare the invisible Higgs [93] and $Z$ decays [69, 96]. While the branching ratio limits are very different,

$$\mathrm{BR}_{Z\to\mathrm{inv}} = 20.00\% \pm 0.06\% \qquad\qquad \mathrm{BR}_{h\to\mathrm{inv}} < 24\% \,, \qquad\qquad (19)$$

the actual partial widths for a decay to dark matter are constrained at very similar levels,

$$\Gamma_{Z\to\chi\chi} < 2 \text{ MeV} \qquad\qquad \Gamma_{h\to\chi\chi} < 1.3 \text{ MeV} \,. \qquad\qquad (20)$$

Table 1: Overview of the constraints on the dark matter sector.

| Observable | Constraint |
|---|---|
| $\Gamma_{Z\to\chi\chi}$ | $< 2$ MeV [69] |
| $\Gamma_{h\to\chi\chi}$ | $< 1.3$ MeV [93] |
| $m_{\tilde{\chi}_1^\pm}$ | $> 103.5$ GeV [69] |
| $\Omega_\chi h^2$ | $0.1187 \pm 20\%$ [1] |
| $\sigma_{\mathrm{SI}}$ | Xenon1T [111], PandaX [113] |
| $\sigma_{\mathrm{SD}}^p$ | Pico60 [115] |
| $\sigma_{\mathrm{SD}}^n$ | LUX [116] |

Throughout our analysis we use the SFITTER tool box [82]. This includes calculating the particle spectrum with SUSPECT3 [97], the Higgs branching ratios with SUSY-HIT and HDE-CAY [99], the relic density and the direct detection rate with MICROMEGAS [102], and the LHC cross sections at leading order with MADGRAPH5 [103]. Higher-order corrections to the LHC cross sections are known [104], but the NLO corrections are typically too small to make a difference to our arguments.

# 4 Final state decays in the MSSM

To estimate the power of mono-*X* analysis from final state decays we need a dark matter model with several particles, where the heavier states have an enhanced production rate at the LHC. Supersymmetric winos and higgsinos are obvious and established candidates for such searches. While for example the bino fraction allows us to explain the relic density with a light neutralino, the winos and higgsinos couple strongly to our SM mediators. We will discuss such signatures first for the MSSM, where we have to negotiate a large LHC rate with the relic density and direct detection (DD) constraints. Ignoring these constraints would allow us to quote much large LHC rates, but we feel that this would mean taking the experimentalists for a ride. Because the main change in the NMSSM electroweakino sector is a new mediator, we can use this extension to estimate an increased LHC reach from non-SM mediators.

## 4.1 Mono-Z

In the MSSM framework, mono-*Z* production is defined as the hard process

$$pp \to \tilde{\chi}_1^0 \tilde{\chi}_1^0 Z \, . \tag{21}$$

As long as we decouple the sfermions and heavy Higgs bosons, the diagrams shown in Fig. 5 are the only diagrams contributing to this process at tree level. This means we can separate three distinct topologies

$$
\begin{aligned}
pp \to ZZ \to Z \, (\tilde{\chi}_1^0 \tilde{\chi}_1^0) && \text{ISR} \\
pp \to Zh \to Z \, (\tilde{\chi}_1^0 \tilde{\chi}_1^0) && \text{invisible Higgs decays} \\
pp \to \tilde{\chi}_j^0 \tilde{\chi}_1^0 \to (\tilde{\chi}_1^0 Z) \, \tilde{\chi}_1^0 && \text{heavy neutralinos } j = 2, 3, 4 \, . 
\end{aligned} \tag{22}
$$

To avoid issues with gauge invariance we always include all topologies in our simulation. If kinematically allowed, intermediate on-shell states lead to a significant enhancement of the LHC production rate in all three cases.

The first two topologies gain impact when the neutralinos are lighter than 45 GeV or 62 GeV. Because of the LEP limits on charginos, this implies that the dark matter agent cannot be a wino or a higgsino and instead requires a sizable bino admixture. Based on the couplings discussed in Sec. 3.1, invisible *Z*-decays require a large higgsino fraction, leading us to

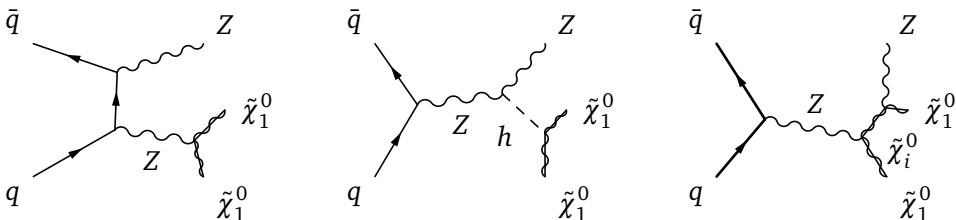

Figure 5: Feynman diagrams contributing to mono-*Z* production in the MSSM, including initial-state *Z*-radiation with a *Z*-portal, *Zh* production with a SM-like Higgs portal, and heavy neutralino decays.

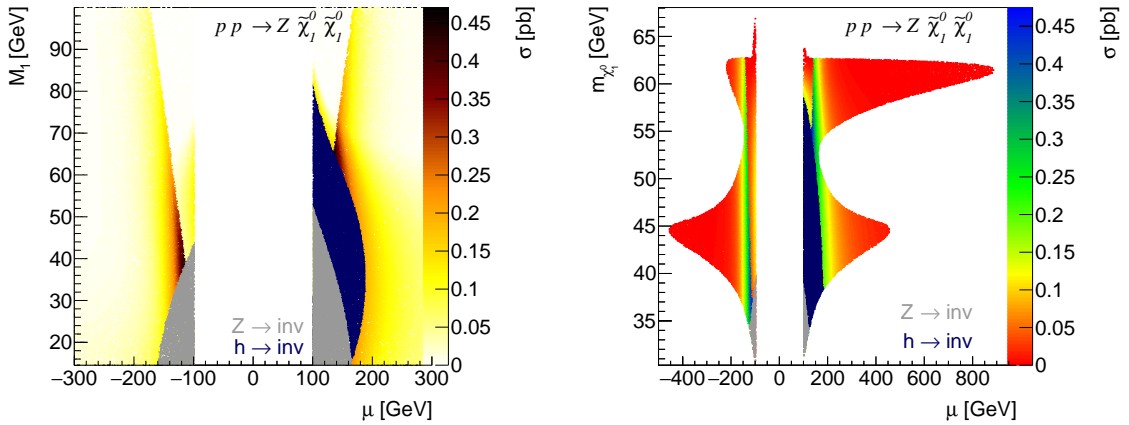

Figure 6: Cross section profiles for the mono-$Z$ process in the $\mu - M_1$ plane. All points fulfill the chargino mass bound (left) and, in addition, predict at most the measured relic density (right). Regions excluded by invisible $Z$ and Higgs decays are shown in light gray and dark blue.

focus on bino–higgsino dark matter. Similarly, invisible SM-like Higgs decays [117] require gaugino–higgsino mixing, or in our case also bino–higgsino dark matter.

For the third topology with its intermediate heavy neutralinos the production process requires a sizable higgsino content in both of the neutralinos involved. The decay $\tilde{\chi}_j^0 \to Z\tilde{\chi}_1^0$ is mediated by the same coupling, giving

$$\sigma_{\tilde{\chi}_1^0 \tilde{\chi}_1^0 Z} \propto \frac{g_{Z\tilde{\chi}_1^0 \tilde{\chi}_j^0}^4}{\Gamma_{\tilde{\chi}_j^0}} \,. \tag{23}$$

It is then crucial that the mass difference between the two relevant neutralinos is large, $m_{\tilde{\chi}_j^0} - m_{\tilde{\chi}_1^0} > m_Z$. For dominantly higgsino dark matter with $m_Z \ll |\mu \pm M_1|, |\mu \pm M_2|$ we can approximate [124]

$$m_{\tilde{\chi}_{1,2}^0} = |\mu| + \frac{m_Z^2(1 \pm s_{2\beta})(\mu \mp M_1 c_w^2 \pm M_2 s_w^2)}{2(\mu \mp M_1)(\mu \mp M_2)}$$

$$m_{\tilde{\chi}_2^0} - m_{\tilde{\chi}_1^0} = m_Z \left( \frac{m_Z}{M_2} c_w^2 + \frac{m_Z}{M_1} s_w^2 \right) \,. \tag{24}$$

This mass difference is always smaller than $m_Z$ [35,126], again indicating that higgsinos alone will not lead to a large mono-$Z$ signal. The obvious solution is to again add a sizable bino content to the dark matter candidate and analyze all three topologies in the limit

$$M_1 < |\mu| \ll M_2 \,, \tag{25}$$

with three propagating neutralinos.

In the left panel of Fig. 6 we show the combined LHC production and decay rate for all three mono-$Z$ topologies in the $\mu - M_1$ plane. The dominant contribution to the sizable rate slightly below the pb range comes from on-shell heavy neutralinos. In the absence of all constraints, the slight asymmetry in the sign of $\mu$ comes from the decay threshold as a function of $\mu$ and $M_1$. Limits from invisible $Z$-decays constrain small $M_1$ values through the dark matter mass and small $|\mu|$ through the higgsino fraction. In contrast, invisible decays of the SM-like Higgs

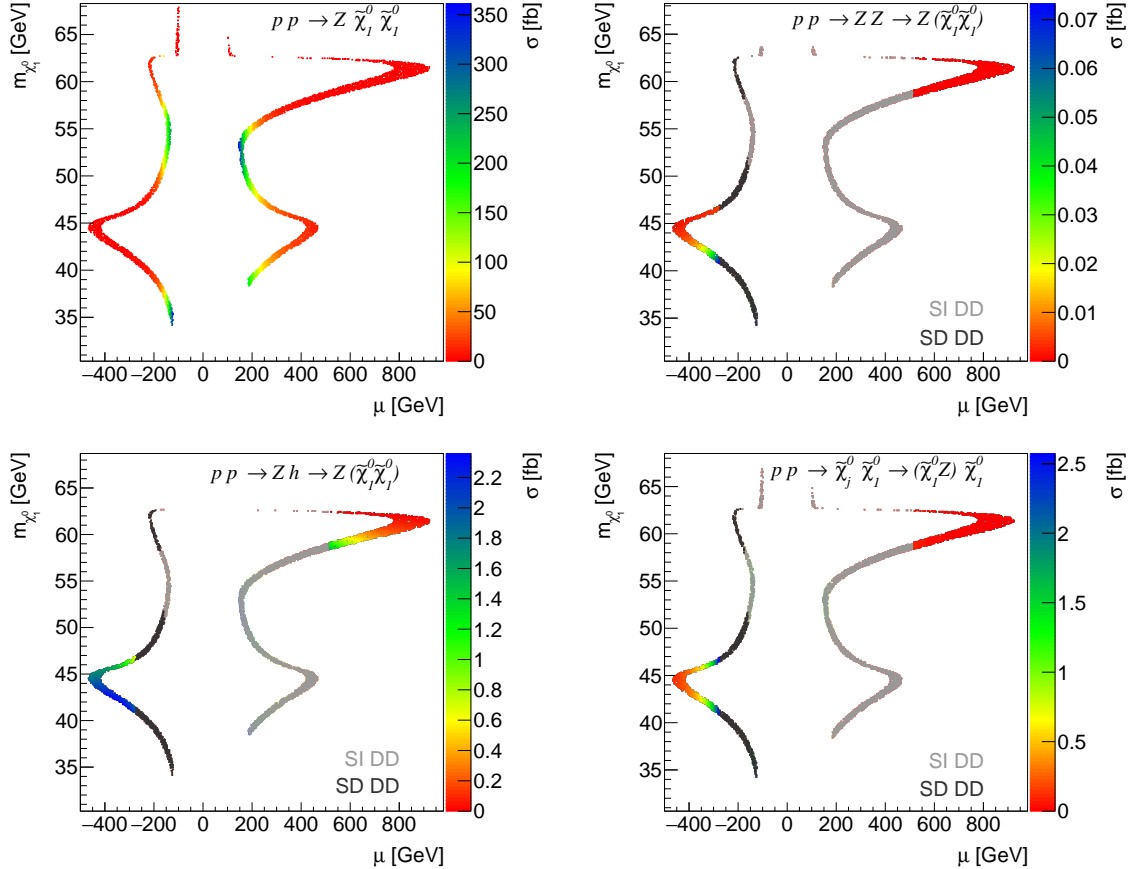

Figure 7: Mono-$Z$ cross sections in agreement with the observed relic density (upper left). For the three LHC topologies we also show the combination with DD limits (upper right to lower right). Points excluded by spin-independent DD limits are light gray, points excluded by spin-dependent direct detection in dark gray.

require a large bino–higgsino mixing and are therefore sensitive to the relative sign of $N_{13}$ and $N_{14}$ in Eq.(10). This leads to a cancellation and hence weaker constraints for $\mu < 0$.

As mentioned above, in any realistic thermal dark matter model the observed relic density is a major constraint. Even the weaker assumption that a given dark matter candidate only contributes a fraction of the observed relic density translates into a relevant lower limit on the dark matter annihilation rate. In the right panel of Fig. 6 we show the allowed parameter space in terms of the dark matter mass $m_{\tilde{\chi}_1^0}$ and $\mu$. The general feature is that for a given dark matter mass the relic density defines minimum coupling strengths for bino–higgsino dark matter, translated into maximum values of $\mu$. The Higgs poles are highly asymmetric with respect to the sign of $\mu$, while the $Z$ poles are approximately symmetric. Because of the on-shell enhancement of the annihilation rate, the invisible decay constraints do not significantly constrain these parameter regions. Other annihilation channels would appear for example for heavier dark matter, but since we are interested in large LHC production rates we limit ourselves to $m_{\tilde{\chi}_1^0} < 70$ GeV at this stage.

In the upper left panel of Fig. 7 we start with all parameter points in agreement with the observed relic density. The curve is identical to the shape shown in Fig. 6. The important result is that for the $Z$ and Higgs funnels the higgsino fractions are relatively small, leading to mono-$Z$ rates around 10 fb at the LHC. Larger LHC rates up to 350 fb are possible, but in

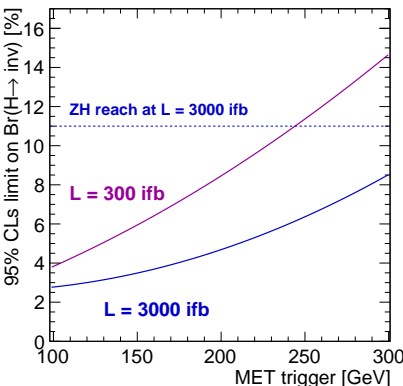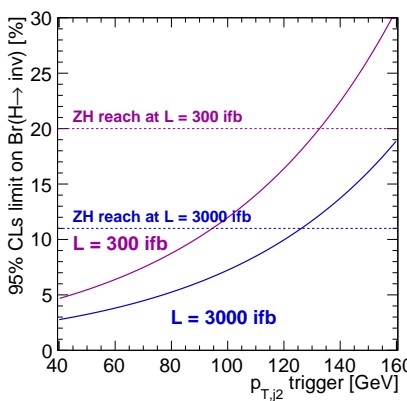

Figure 8: CLs limits on invisible Higgs decays from weak boson fusion, as a function of trigger cuts on missing transverse energy (left) or the transverse momentum of the tagging jets (right), compared to the expected reach in the leptonic $Zh$ channel. Figures from Ref [127].

regions where the dark matter annihilation is not enhanced by on-shell diagrams.

When we include the exact relic density constraint, we should also consider the DD limits displayed in Tab. 1. Based on the spin-independent and spin-dependent interpretations the limits translate into limits on the $g_{Z\tilde{\chi}_1^0\tilde{\chi}_1^0}$ and $g_{h\tilde{\chi}_1^0\tilde{\chi}_1^0}$ couplings, competitive with the full range of the on-shell peaks in Fig. 6. In the remaining three panels of Fig. 7 we show all parameter points predicting the observed relic density and indicate if they agree with the current DD constraints.

The upper right panel of Fig. 7 shows the results for the ISR topology. First, we observe some general features from the interplay of the relic density constraint with spin-independent and spin-dependent direct detection. Just like the shape of the Higgs pole annihilation, the spin-independent constraints are very asymmetric in the sign of $\mu$. This reflects the mixed bino–higgsino coupling to the Higgs with a relative sign between $N_{13}$ and $N_{14}$. Large preferred values of $\mu > 0$ imply small $g_{h\tilde{\chi}_1^0\tilde{\chi}_1^0}$ and correspond to the usual peak in the allowed parameter space. This peak is not (yet) ruled out by direct detection. For $\mu < 0$ the spin-independent DD constraints are weak, so the leading constraints are spin-dependent limits. Even for $m_{\tilde{\chi}_1^0} \approx m_h/2$ they are driven by $g_{Z\tilde{\chi}_1^0\tilde{\chi}_1^0}$.

As expected from our general ISR discussion in Sec. 2, the expected LHC mono-$Z$ rates are very small. They reach 0.07 fb at most, and in a very small region of parameter space around $m_{\tilde{\chi}_1^0} \approx 42$ GeV. This is the only region of parameter space where the LHC process is still enhanced by an on-shell $Z$-decay, but the couplings are not ruled out spin-dependent direct detection.

The next, lower left panel shows the same information for the $Zh$ topology combined with invisible Higgs decays. The structure is similar to ISR case, but with significantly large cross sections. The reason are the limits from invisible $Z$ and Higgs decays, which following Sec. 3.3 look similar in terms of the partial width, but very different in terms of invisible branching ratios. The latter are relevant for the different $(2 \to 2)$ mono-$Z$ channels. Driven by the relic density constraint the largest rate for the $Zh$ topology of around 2 fb appears for $m_{\tilde{\chi}_1^0} \approx 41$ GeV. The large Higgs couplings are barely allowed by DD constraints.

We can skip a dedicated analysis of mono-$Z$ production in the $Zh$ topology and instead resort to the literature [128]: the problem is that we can search for exactly the same model using invisible Higgs decays in weak boson fusion [127, 133]. In Fig. 8 we show the results

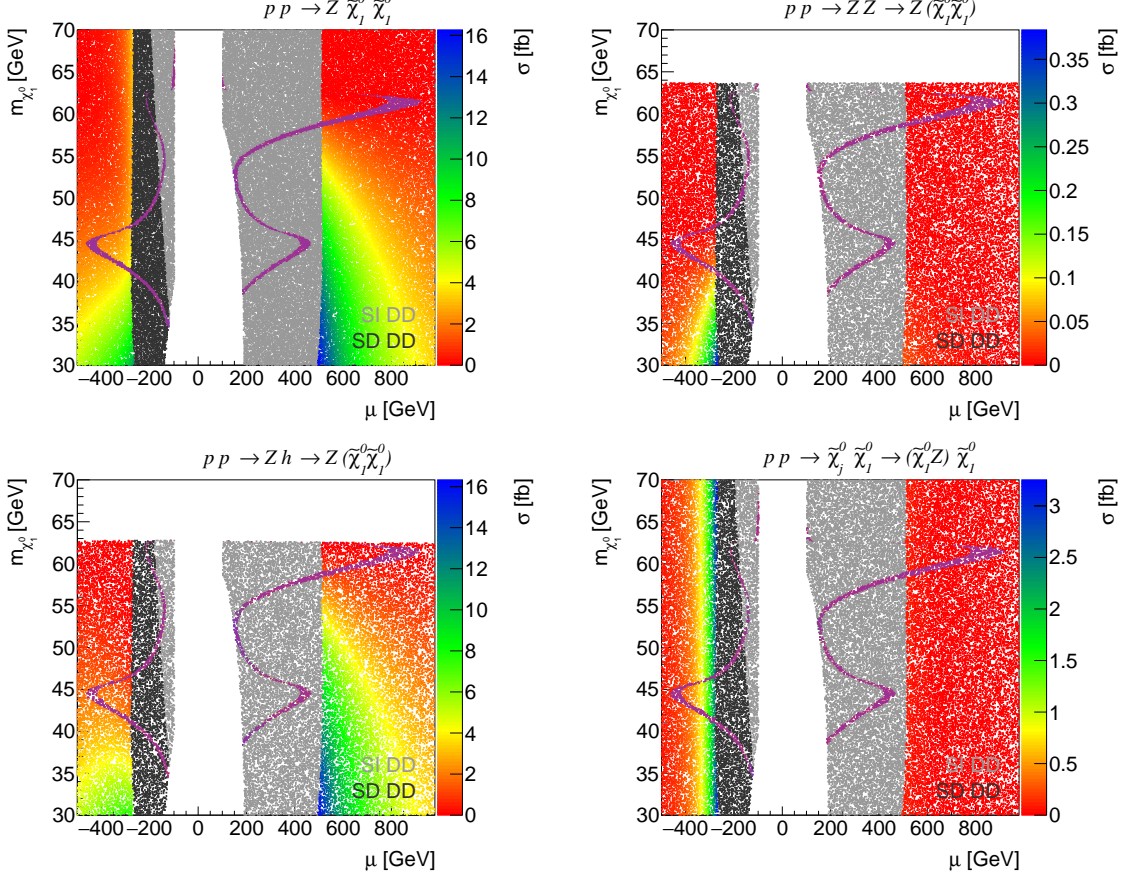

Figure 9: Mono-$Z$ cross sections with DD limits, but without requiring the observed relic density (upper left). Again, we show the three LHC topologies separately (upper right to lower right). Points excluded by spin-independent DD limits are light gray, points excluded by spin-dependent direct detection in dark gray. Points that match the observed relic density under standard assumptions are shown in purple.

from Ref. [127] which indicate that even with conservative assumptions on triggering at the high-luminosity LHC the $Zh$ topology will never be the discovery channel for such dark matter models.

Finally, we show the expected rates for on-shell neutralinos in the lower right panel. Typical mono-$Z$ rates can reach 2.5 fb for light dark matter, $m_{\tilde{\chi}_1^0} = 40 \ldots 47$ GeV. This window is given by the relic density requirement, where annihilation off the $Z$-pole is preferred because of the larger corresponding couplings. While the rate for this topology does not reach the invisible Higgs rates, this channel generally extends to larger dark matter masses. The limiting factor is the lower limits on the heavier two higgsino masses and the corresponding LHC production cross sections through an $s$-channel $Z$.

As a caveat to the discussion above, we note that including the relic density as a constraint relies on assuming a standard cosmological history. In particular, this requires standard thermal DM production and freeze-out. Further, it assumes that the LSP is the only component of dark matter. Since it is not entirely clear that these assumptions hold, we show mono-$Z$ rates including DD limits, but without imposing the relic density constraint, in Fig. 9. Neglecting the relic density allows for significantly larger mono-$Z$ rates. In particular, light DM with mass $m_{\tilde{\chi}_1^0} \lesssim 40$ GeV strongly enhances the decays $Z \to \tilde{\chi}_1^0 \tilde{\chi}_1^0$ and $h \to \tilde{\chi}_1^0 \tilde{\chi}_1^0$. As a consequence, the

$Zh$ topology dominates the mono-$Z$ cross section even more strongly than with the observed relic density and the standard cosmological history imposed.

While we are not arguing that ATLAS and CMS should not perform mono-$Z$ searches, we have seen that any interpretation of such a signal as dark matter is likely to require a modification of the standard thermal freeze-out cosmology. In large parts of the allowed parameter space, the dominant mono-$Z$ topology in the MSSM, after taking into account all constraints, is invisible Higgs decays. Those are best searched for in weak-boson-fusion production [127, 133], while mono-$Z$ production can only confirm the invisible Higgs measurement and add at most very little new information. So there goes the glory of mono-$Z$.

## 4.2 Mono-W(-pairs)

Mono-$W$ production is defined through the hard process

$$pp \to \tilde{\chi}_1^0 \tilde{\chi}_1^0 W^\pm \,. \tag{26}$$

The relevant MSSM diagrams contributing to this process are shown as the first three diagrams in Fig. 10. Like for mono-$Z$ production, we can distinguish three topologies,

$$
\begin{aligned}
pp &\to W^\pm Z \to W^\pm (\tilde{\chi}_1^0 \tilde{\chi}_1^0) && \text{ISR} \\
pp &\to W^\pm h \to W^\pm (\tilde{\chi}_1^0 \tilde{\chi}_1^0) && \text{invisible Higgs decays} \\
pp &\to \tilde{\chi}_j^\pm \tilde{\chi}_1^0 \to (\tilde{\chi}_1^0 W^\pm) \tilde{\chi}_1^0 && \text{heavy charginos } j = 1, 2 \,.
\end{aligned} \tag{27}
$$

The first two rely on the same dark matter couplings as their mono-$Z$ counterparts and only differ in the production process of the SM-like mediators. Therefore, we will again focus on bino-higgsino dark matter for the ISR and invisible Higgs topologies.

Also in analogy to mono-$Z$ production, a third topology features heavy states from the dark matter sector decaying into dark matter and a weak boson. The heavy state is one of the two charginos with the decay $\tilde{\chi}_j^\pm \to W \tilde{\chi}_1^0$. Again, production and decay are mediated by the same coupling,

$$\sigma_{\tilde{\chi}_1^0 \tilde{\chi}_1^0 W} \propto \frac{g_{W\tilde{\chi}_1^0 \tilde{\chi}_j^\pm}^4}{\Gamma_{\tilde{\chi}_j^\pm}} \,. \tag{28}$$

The coupling $g_{W\tilde{\chi}_1^0 \tilde{\chi}_j^\pm}$ is in part a higgsino-higgsino coupling, which following Sec. 4.1 leads us to consider bino-higgsino dark matter. In addition, $g_{W\tilde{\chi}_1^0 \tilde{\chi}_j^\pm}$ includes a wino-wino interaction. However, bino-wino dark matter is difficult to reconcile with LEP bounds in the absence of explicit bino-wino mixing in the neutralino mass matrix. Therefore, all three mono-$W$ topologies again lead us to focus on bino-higgsino dark matter with

$$M_1 < |\mu| \ll M_2 \,, \tag{29}$$

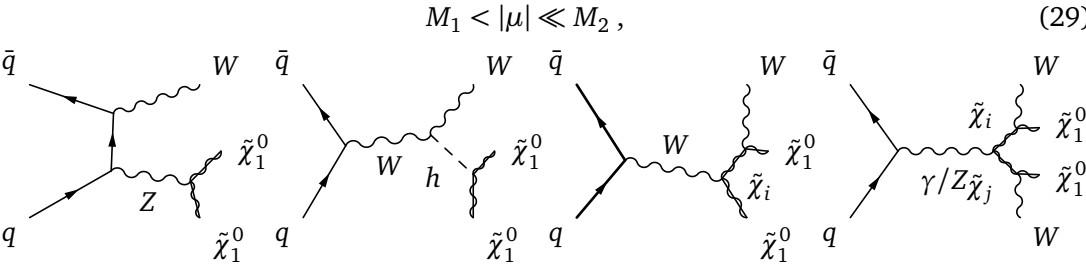

Figure 10: Feynman diagrams contributing to mono-$W$ production in the MSSM, including initial-state $W$-radiation with a $Z$-portal, $Wh$ production with a SM-like Higgs portal, chargino decays, and $W$-pair production.

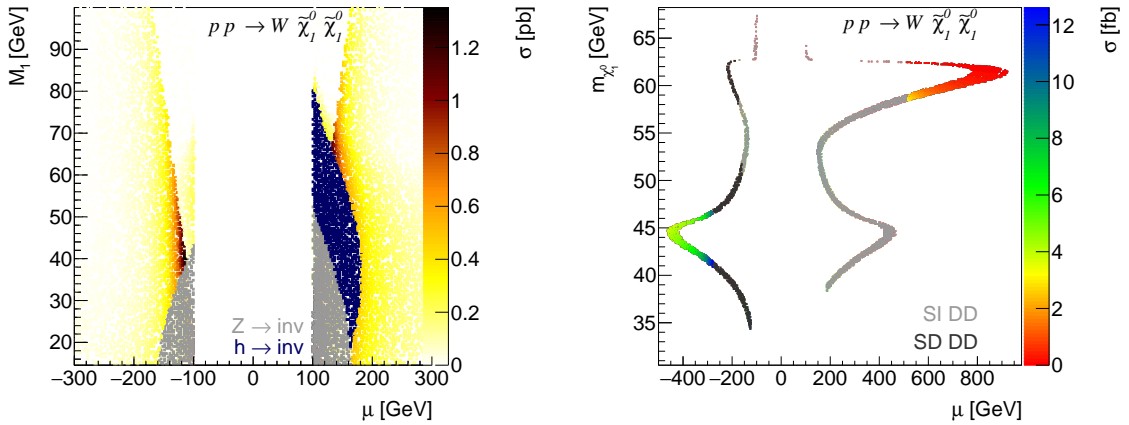

Figure 11: Cross section for the mono-$W$ process in the $\mu - M_1$ plane. Left: points fulfilling the chargino mass bound, shown with the limits on invisible $Z$ and Higgs decays. Right: points also predicting the correct relic density, shown with DD bounds.

just like for the mono-$Z$ analysis in Sec. 4.1.

In the left panel of Fig. 11 we show the mono-$W$ rate in the $\mu - M_1$ plane. Like in Fig. 6 we again include the limits on invisible decays. The largest rates lie in the pb range and stem from the chargino-decay topology. They are two to three times as large as the largest mono-$Z$ rates passing the same constraints. This is due partly to the combination of mono-$W^+$ and mono-$W^-$ production, and partly to the relevant $Z$ and $W$ couplings.

In the right panel of Fig. 11 we show the points in agreement with the observed relic density. In addition, we indicate spin-independent and spin-dependent DD limits. Since the mono-$W$ topologies rely on the same type of dark matter couplings as mono-$Z$ production, the constraints work the same way as in Sec. 4.1: ISR rates become negligible, while rates from chargino decays are suppressed by the large (charged) higgsino masses required by direct detection. The largest LHC rates are again found in a narrow window around the $Z$-pole annihilation funnel. The only difference is that typically mono-$W$ rates are roughly twice as large as mono-$Z$ rates.

A major constraint on mono-$Z$ and mono-$W$ rates at the LHC are DD limits. Both, spin-independent and spin-dependent DD limits impose a strong upper bound on the higgsino admixture in the dark matter candidate through the $g_{Z\tilde{\chi}_1^0\tilde{\chi}_1^0}$ and $g_{h\tilde{\chi}_1^0\tilde{\chi}_1^0}$ couplings. We can try to circumvent them through an LHC production process which survives the limit $N_{13}, N_{14} \to 0$. This happens for mono-$W$-pair production

$$pp \to \tilde{\chi}_i^+ \tilde{\chi}_j^- \to (\tilde{\chi}_1^0 W^+)(\tilde{\chi}_1^0 W^-) \qquad \text{with } i, j = 1, 2 \,, \tag{30}$$

shown in the right diagram of Fig. 10. The rate for chargino pair production through an $s$-channel photon is strongly enhanced compared to purely weak mono-$W$ production. It does not have a counterpart in mono-$Z$ production. Furthermore, even for small couplings we can assume

$$\text{BR}\left(\tilde{\chi}_1^\pm \to W^\pm \tilde{\chi}_1^0\right) \approx 1 \,, \tag{31}$$

since it is the only kinematically allowed two-particle decay mode at tree level. We show the rates for mono-$W$-pair production in Fig. 12. Before taking into account DD constraints,

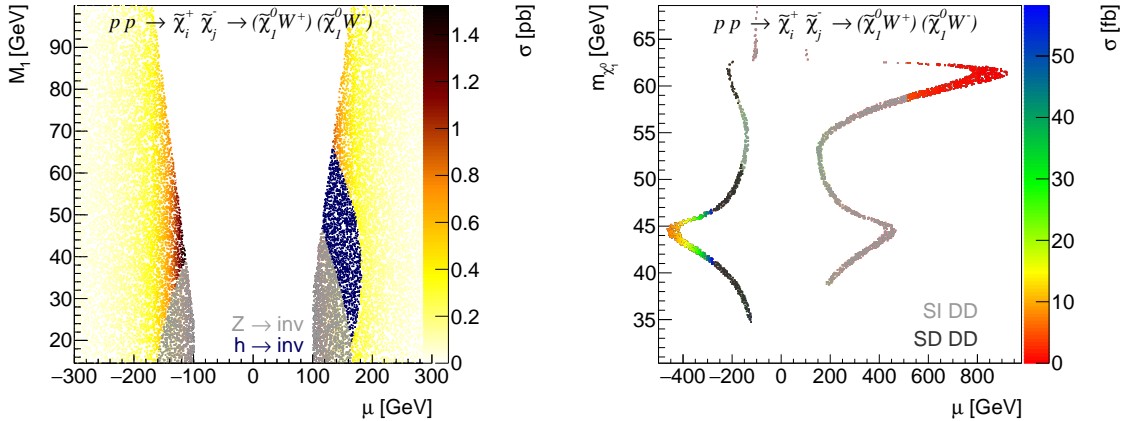

Figure 12: Cross section for the mono-$W$-pair process in the $\mu - M_1$ plane in analogy to Fig. 11. Left: points fulfilling the chargino mass bound, shown with the limits on invisible $Z$ and Higgs decays. Right: points also predicting the correct relic density, shown with DD bounds.

the rates for mono-$W$ and mono-$W$-pair production are similar. Since the actual couplings are not constrained by direct detection, the maximum rates remain larger than for mono-$W$ production. However, we find that the spin-dependent DD bound on the neutral higgsino, $|\mu| \gtrsim 250$ GeV leads to a kinematic suppression of the $\tilde{\chi}_j^- \tilde{\chi}_j^+$ production rate.

Our mono-$W$ study implies that in contrast to, for instance, effective theory arguments, intermediate on-shell states prefer mono-$W$ production over mono-$Z$ production. One of the mechanisms behind this is the mono-$W$-pair topology. Its contributions are removed, if we employ jet or lepton vetoes to remove top backgrounds for the mono-$W$ signal. Again, there is no point in performing a detailed signal-background analysis of this channel, because chargino pair production is a bread-and-butter signature for electroweakinos at the LHC [134].

## 4.3 Mono-Higgs(-pairs)

Mono-Higgs production is the third electroweak process we consider in our comprehensive study of final state decay leading to mono-$X$ signatures. The hard process reads

$$pp \to \tilde{\chi}_1^0 \tilde{\chi}_1^0 h \,. \tag{32}$$

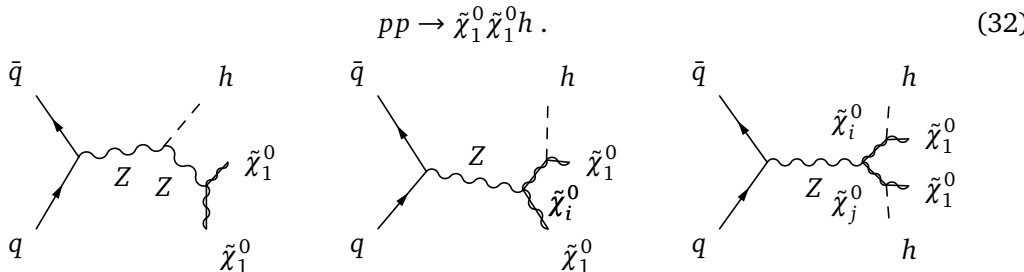

Figure 13: Feynman diagrams contributing to mono-Higgs production in the MSSM, $Zh$ production with a $Z$-portal, and heavy neutralino decays, and Higgs pair production.

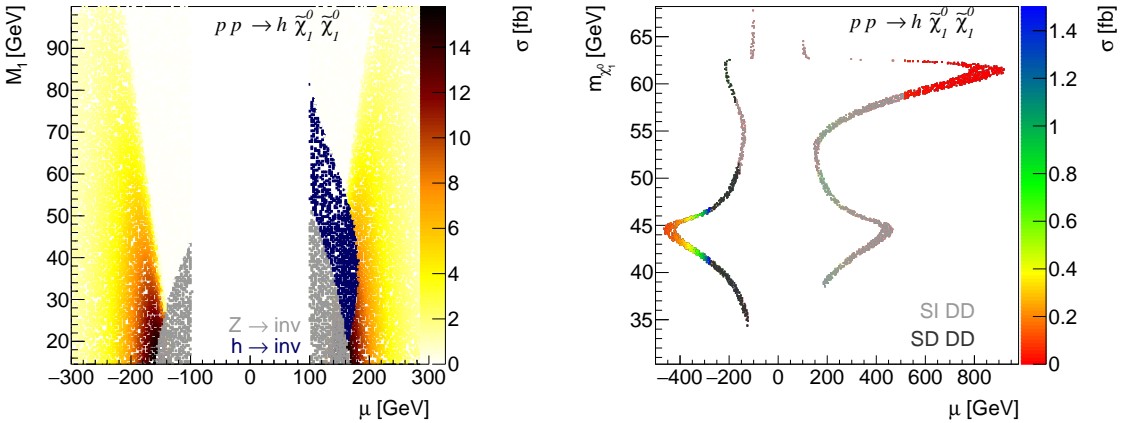

Figure 14: Cross section for the mono-Higgs process in the $\mu-M_1$ plane. Left: points fulfilling the chargino mass bound, shown with the limits on invisible $Z$ and Higgs decays. Right: points also predicting the correct relic density, shown with DD bounds.

The Higgs boson in the final state is the SM-like light scalar of the MSSM. The Feynman diagrams shown in Fig. 13 define two mono-Higgs topologies

$$pp \rightarrow hZ \rightarrow h\,(\tilde{\chi}_1^0 \tilde{\chi}_1^0) \qquad\qquad \text{invisible } Z\text{-decays}$$

$$pp \rightarrow \tilde{\chi}_j^0 \tilde{\chi}_1^0 \rightarrow (\tilde{\chi}_1^0 h)\,\tilde{\chi}_1^0 \qquad\qquad \text{heavy neutralinos } j = 2,3,4\,. \tag{33}$$

Obviously, the usual ISR topology is not relevant for the Higgs case. The $Zh$ topology is based on the same production mechanism as for mono-$Z$ production, but combined with a strongly constrained branching ratio $\text{BR}_{Z\rightarrow\chi\chi}$. The two relevant couplings driving the neutralino decay topology are

$$\sigma_{\tilde{\chi}_1^0 \tilde{\chi}_1^0 h} \propto \frac{g_{Z\tilde{\chi}_1^0 \tilde{\chi}_i^0}^2 g_{h\tilde{\chi}_1^0 \tilde{\chi}_i^0}^2}{\Gamma_{\tilde{\chi}_i^0}}\,. \tag{34}$$

The production process still requires a sizable coupling to the $Z$, while the decay proceeds through the Higgs coupling. The decay $\tilde{\chi}_i^0 \rightarrow \tilde{\chi}_1^0 h$ competes with the decay $\tilde{\chi}_i^0 \rightarrow \tilde{\chi}_1^0 Z$. Just like for mono-$Z$ and mono-$W$ production, the observed relic density combined with all available constraints motivates mixed bino-higgsino dark matter,

$$M_1 < |\mu| \ll M_2\,. \tag{35}$$

In the left panel of Fig. 14 we show the rates we start with, before considering relic density and DD constraints. We see that the mono-Higgs rates are more than an order of magnitude smaller than their mono-$Z$ or mono-$W$ counterparts shown in Fig. 6 and Fig. 11. For the $Zh$ topology the limiting factor is the smaller invisible branching ratio of the $Z$-boson as compared to the invisible Higgs decays, described in Sec. 3.3. The neutralino decay topology predicts smaller rates because especially for large production rates and before considering DD limits the competing decay rate $\tilde{\chi}_{2,3}^0 \rightarrow \tilde{\chi}_1^0 Z$ is large.

In the right panel of Fig. 14 we see the effect of the spin-dependent and spin-independent DD limits. The $Zh$ topology is now suppressed to unobservable LHC rates through the invisible $Z$ branching ratio, just like the ISR topology of the mono-$Z$ signature described in Sec. 4.1. Unlike for the mono-$Z$ case, the neutralino decay topology becomes the leading channel with possible LHC rates in the range of 1 fb. The predicted mono-Higgs rate after taking into account

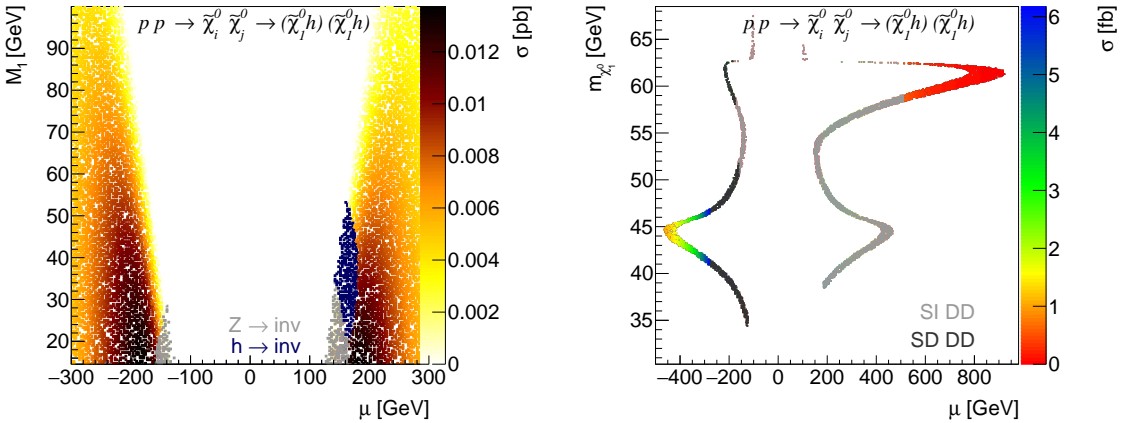

Figure 15: Cross section for the mono-Higgs-pair process in the $\mu - M_1$ plane. Left: points fulfilling the chargino mass bound, shown with the limits on invisible $Z$ and Higgs decays. Right: points also predicting the correct relic density, shown with DD bounds.

DD constraints is indeed not much smaller than the expected mono-$Z$ rates from neutralino decay.

Inspired by the mono-$W$ case, it turns out that one way out of some of the leading constraints is mono-Higgs-pair production shown in the right panel of Fig. 13,

$$pp \rightarrow \tilde{\chi}_i^0 \tilde{\chi}_j^0 \rightarrow (\tilde{\chi}_1^0 h)(\tilde{\chi}_1^0 h) \qquad \text{with } i, j = 2, 3, 4 \,. \tag{36}$$

The neutralino production couplings are now separated from the decay couplings and, more importantly, from the couplings mediating direct detection,

$$\sigma_{\tilde{\chi}_1^0 \tilde{\chi}_1^0 hh} \propto \frac{g_{Z\tilde{\chi}_i^0 \tilde{\chi}_j^0}^2 g_{h\tilde{\chi}_1^0 \tilde{\chi}_i^0}^2 g_{h\tilde{\chi}_1^0 \tilde{\chi}_j^0}^2}{\Gamma_{\tilde{\chi}_i^0} \Gamma_{\tilde{\chi}_j^0}} \,. \tag{37}$$

In our preferred scenario with bino-higgsino dark matter and another, relatively light higgsino the production of heavy neutralino pairs will be sizable. At the same time, the decay to Higgs bosons requires a gaugino content just like the annihilation responsible for the correct relic density.

We show the LHC rates for the mono-Higgs-pair signature in Fig. 15. First, the mono-Higgs-pair cross section is suppressed by the phase space of two heavy higgsinos in the final state with $|\mu| \gtrsim 300 \dots 400$ GeV, just like the mono-$W$-pair rate. This is why the rate before applying any constraints is in the same range as the mono-Higgs rate. On the other hand, every coupling contributing to the LHC rate is unrelated to direct detection. Through a large bino fraction of the dark matter agent we can essentially decouple the DD constraints, so the LHC rates with and without relic density and DD constraints are very similar. All we need to do is enhance the annihilation rate in the early universe through an on-shell condition $m_{\tilde{\chi}_1^0} \approx M_1 \approx m_Z/2$ or $m_{\tilde{\chi}_1^0} \approx M_1 \approx m_h/2$.

The LHC signature of mono-Higgs-pair production is similar to Higgs pair production at the LHC. While the expected production rate for a pair of SM-like Higgs bosons is around 35 fb, the additional missing energy in the mono-Higgs-pair signal of Eq.(36) should allow for a better background rejection. Which decay combination of the two Higgs bosons works best

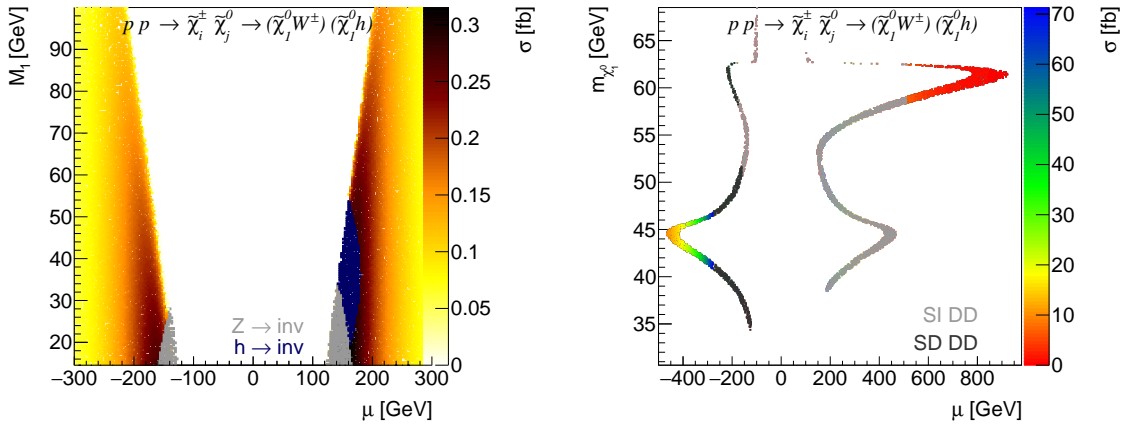

Figure 16: Cross section for the mono-$Wh$-pair process in the $\mu - M_1$ plane. Left: points fulfilling the chargino mass bound, shown with the limits on invisible $Z$ and Higgs decays. Right: points also predicting the correct relic density, shown with DD bounds.

for this purpose is currently under study [137]. For SM-like Higgs pairs the combination $b\bar{b}\,\gamma\gamma$ works best to guarantee detection and reduce backgrounds, but for the smaller dark matter signal the combinations $b\bar{b}\,b\bar{b}\,\not{E}_T$ or $b\bar{b}\,WW\,\not{E}_T$ might be more promising.

Finally, for completeness and in analogy to mono-$W$ pairs and mono-Higgs pairs, we consider the mono-$Wh$ pair process given by

$$pp \to \tilde{\chi}_i^{\pm} \tilde{\chi}_j^0 \to (\tilde{\chi}_1^0 W^{\pm})(\tilde{\chi}_1^0 h) \qquad \text{with } i = 1, 2 \text{ and } j = 2, 3, 4. \tag{38}$$

This topology is driven by the production coupling $g_{W\tilde{\chi}_j^0\tilde{\chi}_i^{\pm}}$ and, for the decay, $g_{W\tilde{\chi}_1^0\tilde{\chi}_i^{\pm}}$ and $g_{h\tilde{\chi}_1^0\tilde{\chi}_j^0}$,

$$\sigma_{\tilde{\chi}_1^0\tilde{\chi}_1^0 Wh} \propto \frac{g^2_{W\tilde{\chi}_j^0\tilde{\chi}_i^{\pm}} g^2_{W\tilde{\chi}_1^0\tilde{\chi}_i^{\pm}} g^2_{h\tilde{\chi}_1^0\tilde{\chi}_j^0}}{\Gamma_{\tilde{\chi}_i^{\pm}}\Gamma_{\tilde{\chi}_j^0}}. \tag{39}$$

In the scenario of a bino-higgsino LSP, heavier higgsinos and a decoupled wino, the production cross section for a heavy chargino-neutralino pair will be sizable. Like in the mono-$W$ (-pair) process, we again have

$$\text{BR}\left(\tilde{\chi}_1^{\pm} \to W^{\pm}\tilde{\chi}_1^0\right) \approx 1. \tag{40}$$

On the other hand, the decay $\tilde{\chi}_j^0 \to \tilde{\chi}_1^0 h$ of the heavy neutralino, requiring gaugino and higgsino parts in the LSP, competes with the decay $\tilde{\chi}_j^0 \to \tilde{\chi}_1^0 h$.

Hence, rates before relic density and direct detection constraints, shown in the left panel of Fig. 16, lie between the rates for mono-Higgs pairs and those for mono-$W$ pairs. Like for mono-$W$ pairs and mono-Higgs pairs, production couplings and decays are decoupled from direct detection. The correct relic density can be guaranteed through the resonant enhancement at $m_{\tilde{\chi}_1^0} \approx M_1 \approx m_Z/2$ or $m_{\tilde{\chi}_1^0} \approx M_1 \approx m_h/2$. Hence, the mono-$Wh$-pair cross section is only suppressed kinematically through the production of heavy higgsinos. The resulting rates are shown in the right panel of Fig. 16. We find cross sections of up to 70 fb, slightly above the mono-$W$-pair rate. We do not perform a signal-background analysis, since neutralino-chargino pairs belong to the electroweakino signatures already being studied at the LHC [136].

# 5 Final state decays beyond the MSSM

The leading constraint on the size of electroweak mono-$X$ signals in the MSSM comes from direct detection or, more specifically, from the combination of the relic density constraint and the DD limits. The reason is that we need large couplings to the $Z$ and SM-like Higgs mediators to reach the observed relic density, direct detection strongly constrains these couplings, and most LHC rates again rely on the same couplings. In extended models like the NMSSM a dark sector mediator is responsible for the correct relic density, in spite of very small couplings to the Standard Model. From our mono-$W$(-pair) and mono-Higgs(-pair) we know how to decouple the decay topologies at the LHC from the DD constraints, which motivates our NMSSM study.

Following Sec. 3.2 we adjust the singlet–singlino dark matter sector such that a light singlino with $m_{\tilde{\chi}_1^0} = 10$ GeV can annihilate to the correct relic density through an on-shell singlet. Because this annihilation relies on the couplings within the singlet–singlino sector we can decouple the gaugino masses in our $|\mu| \ll M_1 = M_2 = 1$ TeV. Following Eq.(14) and Eq.(15) we ensure the corresponding mass relation by choosing $\tilde{\kappa}$ such that

$$m_{\tilde{\chi}_1^0} \approx 2\tilde{\kappa}\mu + \frac{m_Z^2}{\mu} \tilde{\lambda}^2 \frac{2\tilde{\kappa} - s_{2\beta}}{4\tilde{\kappa}^2 - 1} = 10 \text{ GeV} . \tag{41}$$

If we include the LEP constraint $|\mu| \gtrsim 100$ GeV, this typically implies

$$|\tilde{\kappa}| = \frac{m_{\tilde{\chi}_1^0}}{2|\mu|} \lesssim 0.05 . \tag{42}$$

or $|\kappa| \ll |\lambda|$ in the original notation. For our mass hierarchy this means

$$|\tilde{\kappa}\mu| \ll |\mu| \ll M_1 \approx M_2 . \tag{43}$$

The singlino couplings from Eq.(18) are approximately given by

$$g_{a_s \tilde{\chi}_1^0 \tilde{\chi}_1^0} \approx g_{h_s \tilde{\chi}_1^0 \tilde{\chi}_1^0} \approx -\sqrt{2} g \, \tilde{\lambda}\tilde{\kappa} \, N_{15}^2 . \tag{44}$$

They are not large compared for example to gauge couplings, but sufficiently large to explain the observed relic density for an on-shell annihilation process. The remaining free parameters in the NMSSM electroweakino sector which we vary in our analysis are $\tilde{\kappa}$, $\tilde{\lambda}$ and $A_\kappa$.

While it is generally possible to extend all MSSM analyses of Sec. 4 to the NMSSM we focus on the two most interesting cases, the strongly constrained mono-$Z$ signal and the most flexible mono-Higgs-pair signal

$$pp \to \tilde{\chi}_1^0 \tilde{\chi}_1^0 Z \qquad \text{and} \qquad pp \to \tilde{\chi}_i^0 \tilde{\chi}_j^0 \to (\tilde{\chi}_1^0 h)(\tilde{\chi}_1^0 h) . \tag{45}$$

For the mono-$Z$ signal the ISR, invisible SM-like Higgs $h_{125}$, and heavy neutralino topologies shown in Fig. 5 are supplemented by the associated $Zh_s$ mediator production.

As usual, we start with the cross sections without the dark matter constraints in Fig. 17. Because we fix the dark matter mass to 10 GeV, there is no threshold left to consider. Instead, we show the correlation between the higgsino mass and the singlino–higgsino mixing parameter $\tilde{\lambda}$. In general, the LHC cross section grows with $\lambda$, since all contributing diagrams are driven by bino-higgsino mixing, times $\tilde{\lambda}$ connecting the higgsino content to the singlino content. For heavy gaugino masses, the $Z$ and Higgs decay constraints limit the size of the higgsino fraction of the lightest neutralino, or $\tilde{\lambda}$ for a given value of $\mu$. While in the MSSM the invisible Higgs

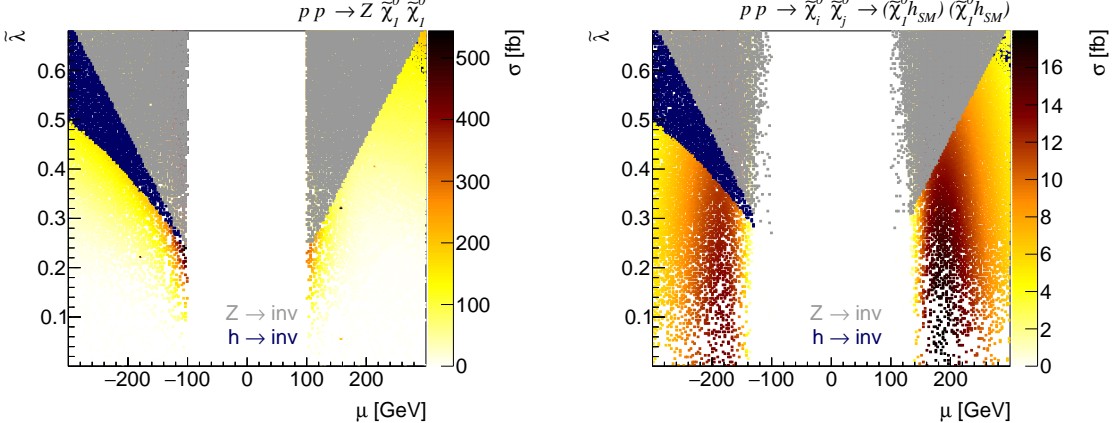

Figure 17: Cross section profiles for the mono-$Z$ (left) and mono-Higgs-pair (right) processes in the $\mu - \lambda$ plane. All points fulfill the chargino mass bounds. Regions excluded by invisible $Z$ and Higgs decays are shown in light gray and dark blue.

limits were stronger for $\mu > 0$, they now constrain mostly $\mu < 0$. This is because of the sign difference between the singlino–Higgs coupling and the bino-Higgs coupling,

$$
g_{h_{125}\tilde{\chi}_1^0\tilde{\chi}_1^0} \approx \begin{cases} g' N_{11} s_\beta \left( -\dfrac{N_{13}}{t_\beta} + N_{14} \right) & \text{bino} \\[3mm] -\sqrt{2}\lambda N_{15}\left( N_{13} + \dfrac{N_{14}}{t_\beta} \right) & \text{singlino.} \end{cases} \tag{46}
$$

The mono-Higgs-pairs rate shown in the right panel of Fig. 17 are similar to the NMSSM case shown in Fig. 15. As expected from the enhanced flexibility in all couplings, they prefer a small higgsino mass and can exceed the mono-$Z$ rates.

The interesting question is, how these large LHC rates change when we apply the constraints from the relic density and direct detection. In the left panel of Fig. 18 we show the results for mono-$Z$ production in the NMSSM framework. The general pattern confirms that either the scalar or the pseudo-scalar mediator has to be just slightly off its mass shell, with a width given by the velocity distribution. The main difference between them arises from CMB bounds, which are irrelevant for scalar p-wave annihilation, while a 10 GeV neutralino is barely allowed for s-wave annihilation through the pseudoscalar. In addition, following Eq.(46) the LHC production rate is roughly proportional to a factor $\lambda$ from the explicit couplings and another factor $\tilde{\lambda}$ from the higgsino fractions.

After including all constraints, the $Zh$ topology with an invisible Higgs again emerges as the dominant mono-$Z$ process. However, while for the MSSM the direct detection constraints effectively enforce $\mathrm{BR}\left(h_{125} \to \tilde{\chi}_1^0\tilde{\chi}_1^0\right) \lesssim 0.003$, they now fall behind the LHC limit of 24%. This way, the LHC rate in the NMSSM can be forty times as large as in the MSSM, exceeding 100 fb. The light, new scalar mediator also leads to spin-independent singlino–nucleon scattering. This manifests itself in the excluded points at low $m_{h_s}$ and large singlino–higgsino mixing $\lambda$.

Also in Fig. 18 we show the same effects for mono-Higgs-pair production. In that case the relevant third parameter is not the singlino–higgsino mixing, but the higgsino mass parameter. In the NMSSM the LHC rates can be three times as large as in the MSSM. The reason is a kinematic effect, because the weaker DD bounds for smaller dark matter masses allow for a larger higgsino fraction in the dark matter candidate and hence lighter on-shell higgsinos. The

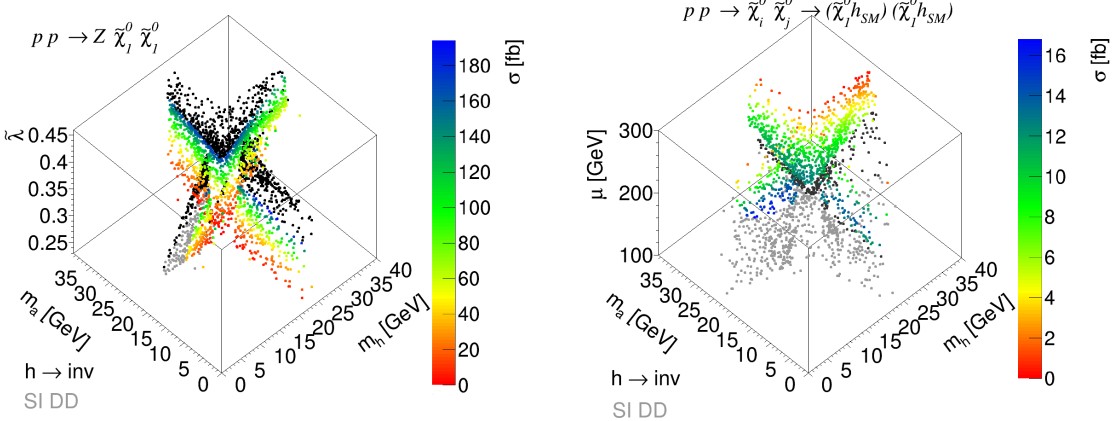

Figure 18: Mono-$Z$ (left) and mono-Higgs-pair (right) cross section versus the singlet-like pseudoscalar and scalar masses and the singlino–higgsino mixing parameter. All points fulfill the relic density, chargino mass, and invisible $Z$ decay bounds. The effects of the invisible Higgs decays and spin-independent direct detection are shown in grey.

subsequent branching ratios for the decays $\tilde{\chi}^0_{2,3} \to h_{125}\tilde{\chi}^0_1$ are similar to typical MSSM values, namely around 40%. For constant $\mu$ this branching ratio is approximately independent of $\lambda$, since both $g_{h\tilde{\chi}^0_1\tilde{\chi}^0_1}$ and $g_{Z\tilde{\chi}^0_1\tilde{\chi}^0_1}$ are proportional to $\lambda^2$ from the explicit and implicit dependences.

Altogether, we indeed see how the light NMSSM mediators allow us to decouple the different relic density, direct detection, and LHC observables. Most importantly, our dark matter singlet as well as the heavier higgsinos can now be lighter than in the MSSM. For all channels this directly translates into an increase of the LHC rate by a factor three to forty. The mono-$W$ channel will obviously follow the same pattern.

# 6 Summary

Mono-$X$ searches are promising strategies to search for dark matter at the LHC. A wealth of models motivate a large number of analyses, including mono-jet, mono-photon, mono-$Z$, mono-$W$, and mono-Higgs searches. Major support for these searches comes from effective theories of dark matter, but with an approach-specific ranking of the different mono-$X$ signatures. In this paper we have compared different mono-$X$ searches based on two orthogonal theoretical assumptions concerning their LHC production.

As a starting point, we have compared mono-jet, mono-photon, and mono-$Z$ searches for for a dark matter toy model with a $Z'$ mediator. In that case all mono-$X$ searches rely on the same ISR topology and can be compared directly. We confirmed that mono-jet searches are by far the most promising, as long as the systematic uncertainties are under control. Combining different ISR-based mono-$X$ searches does not add a significant amount of information, so we can skip ISR signatures for the remaining analysis.

In the main part of the paper we have analyzed decays of heavier states of an extended dark matter sector as a source of electroweak mono-$X$ signals. For electroweakinos in the MSSM we have shown how different intermediate on-shell states lead to large predicted LHC rates, clearly separating our topologies from any effective theory description. Adding relic density and direct detection constraints, we found that mono-$Z$ rates at the LHC cannot be large, if

we stick to SM-like mediators. A leading channel turned out to be associated *Zh* production with an invisible Higgs decay. This associated production signature is known to be weaker than searching for the same invisible Higgs decay in weak boson fusion.

Unlike for effective theories, we found that mono-*W* searches are more promising than mono-*Z* searches, also due to a mono-*W*-pair topology. Experimentally, we need to ensure that the contributing *W*-pair topology is not removed by lepton or jet vetoes as part of the mono-*W* analysis. Initially, mono-Higgs production looks distinctly un-promising, but once we include direct detection constraints the combination of mono-Higgs and mono-Higgs-pair topologies allows us to separate the LHC signal from the particles and couplings driving dark matter annihilation and direct detection.

The NMSSM with its light scalar and pseudo-scalar mediators decouples the relic density, direct detection and LHC rates especially for a light dark matter. In that sense, its extended scalar sector comes much closer to the anything-goes philosophy of simplified models. Correspondingly, the mono-*Z*, mono-*W*(-pair), and mono-Higgs-pair rates at the LHC can be much larger than in the MSSM. Our two leading effects, namely that mono-*W* production is at least as attractive as mono-*Z* production and that mono-*W*-pair and mono-Higgs-pair topologies should be included, remain.

### Acknowledgments

We would like to thank Martin Bauer for many fruitful and fun discussions. The authors acknowledge support by the state of Baden-Württemberg through bwHPC. TP is supported by the DFG Forschergruppe *New Physics at the LHC* (FOR 2239).

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
