# Peer review of "Actual Physics behind Mono-X"

_SciPost Physics, doi:SciPost Phys. 5, 034 (2018)_

## Round 2 · Referee Report · Tim Tait · 2018-7-4

Strengths
1. This article does a comprehensive analysis of the mono-X family of signature, and also applies them to MSSM and NMSSM specific models.
2. The collider analysis is rigorous and well handled and the conclusions are well explained with a mixture of analytical reasoning illustrating numerical results.
3. The results are interesting and suggest some new search strategies such as mono-W-pair which look quite powerful.
Weaknesses
1. Throughout, the authors are somewhat guilty of the very common sin of overly fetishizing the freeze-out relic abundance. That is a nice picture for where the amount of WIMP dark matter comes from, but modifications to cosmology or additional particle physics could over-turn it.
Report
In general, this is a valuable contribution to the literature.
Requested changes
1. This is optional, but I would like to see versions of the SFITTER plots without the relic density imposed. (And of course, also to keep the current versions with it imposed).
Author: Tilman Plehn on 2018-09-24 [id 319]
(in reply to Report 1 by Tim Tait on 2018-07-04)
Hey Tim.
Thank for pointing out this bias. We have now added SFitter plots of mono-Z rates without the relic density imposed and added a short discussion to Sec. 4.1.
Author: Tilman Plehn on 2018-09-24 [id 320]
(in reply to Report 2 on 2018-08-01)First, the direct detection reference were indeed wrong, we have changed the sentence to 'DD limits displayed in Tab. 1'. Second, as also suggested by Tim Tait we have added plots of mono-Z rates without the relic density, as well as a short discussion to Sec. 4.1.

---

## Round 2 · Referee Report · Anonymous · 2018-8-1

Strengths
Serious analysis for collider and astronomical dark matter searches for motivated frameworks beyond the standard model.
Weaknesses
Originality: the theoretical frameworks considered in this work have been studied extensively, also for many of the constraints presented here.
Report
This paper considers mono-X signatures arising from non minimal dark sectors. After a review and comparison of mono-X searches from initial state radiation, as it is the case in effective theories for dark matter, the authors describe the theoretical framework in section 3. They consider two cases: the minimal supersymmetric standard model (MSSN) and the next to the minimal supersymmetric standard model (NMSSN). In the same section, they also present the SFITTER framework, which is the setup for the following analysis. The central results are in sections 4 and 5, where the authors combine mono-X searches (with X = Z, W and Higgs) with astrophysical constraints such as relic density and direct and indirect detection. In the former section they study the MSSN, in the latter the NMSSM. Conclusions are given in section 6.
Requested changes
I find this paper clearly written and of good quality. Before recommending its publication, I am asking for two minor revisions. The authors say that direct detection constraints are derived as described in section 3.3. Unless I am missing something, I do not see direct detection discussed in section 3.3. Finally, my last comment is perhaps obvious to the authors, but I think it is important to emphasize this point. It is not fair to treat collider and relic density bounds on an equal footing. As they correctly state in the paper, the lightest stable particle in their framework could be a subdominant contribution to the dark matter and therefore the relic density is just a lower bound on the cross section. More importantly, the relic density calculations performed in this work are based on the assumption of a standard cosmological history. There is no evidence for this, and motivated beyond the standard model frameworks (including moduli in supersymmetric theories) predict deviation from such a simple picture. It is possible to reproduce the observed dark matter relic density for cross sections much larger or much smaller than the WIMP miracle value. While this is a trivial point, I think it is important that the authors state very clearly this caveat.

---

## Round 3 · Referee Report · Tim Tait (Referee 1) · 2018-9-29

Report

The article looks complete and correct to me.

---

## Round 3 · Referee Report · Anonymous (Referee 2) · 2018-10-1

Report

I am happy with the new version of the paper and I recommend it to be published.

---

## Round 3 · Author Response

We have accommodated all changes suggested by the referees.

---

## Round 3 · List of Changes

Discussion of limits without relic density constraint, mono-Wh channel added below Eq.(38).

---

## Editorial Decision

published